# CXCL13 as a Prognostic Biomarker of Survival Outcomes in Muscle-Invasive Bladder Cancer

## Abstract

Muscle-invasive bladder cancer (MIBC) is associated with poor survival despite advances in therapy. Reliable biomarkers to guide prognosis and stratify patients for therapy remain an unmet need. Here, we evaluate the chemokine CXCL13 as a prognostic factor in MIBC using an AI-assisted hypothesis generation and validation pipeline. Based on prior biological evidence linking CXCL13 to immune activity, we hypothesized that higher CXCL13 expression would be associated with improved survival outcomes.

We analyzed the TCGA-BLCA cohort and stratified patients into CXCL13-high and CXCL13-low groups. Kaplan–Meier survival analysis demonstrated that high CXCL13 expression was significantly associated with improved overall survival (OS; $p = 0.0059$) and progression-free survival (PFS; $p = 0.035$).

These findings support CXCL13 expression as a prognostic biomarker in MIBC and highlight its potential to refine patient risk stratification. More broadly, this study illustrates how AI-generated hypotheses can be systematically validated with open data and human-in-the-loop oversight, ensuring reproducibility of both code and interpretation. Our work underscores the promise of AI-assisted biomarker discovery in oncology.

## 1   Introduction

Muscle-invasive bladder cancer (MIBC) is an aggressive disease with a 5-year survival rate of only ∼40–60% despite multimodal therapy Uysal et al. [2021]. Standard prognosticators—tumor stage, grade, and a few molecular markers—offer limited guidance on treatment decisions. Immune checkpoint inhibitors have improved outcomes in a subset of bladder cancer patients, yet most do not respond Sharma et al. [2017]. A pressing challenge is identifying biomarkers that can reliably predict prognosis and stratify patients who may benefit from therapy.

Currently, PD-L1 expression by immunohistochemistry has been explored, but its predictive value in bladder cancer is inconsistent Aggen and Drake [2017]. Many patients with PD-L1–negative tumors still respond to checkpoint blockade, and vice versa. Tumor mutational burden (TMB) and related mutational signatures (e.g., APOBEC-driven mutations) indicate neoantigen load, yet alone have insufficient specificity in MIBC Robertson et al. [2017]. Emerging multi-gene expression profiles, such as an 18-gene "T cell–inflamed" signature, have shown promise in capturing immune activation more comprehensively Ayers et al. [2017]. However, there remains a need for biomarkers that reflect immune activity and are reproducible across datasets.

CXCL13 is a B-cell chemoattractant chemokine integrally involved in lymphoid neogenesis and the organization of tertiary lymphoid structures (TLS) within tumors Lin et al. [2025]. Recent studies in bladder cancer and other tumor types have linked the presence of TLS—organized lymphoid aggregates in tumors—with improved patient prognosis and enhanced anti-tumor immunity Lin et al. [2025], Cabrita et al. [2020]. In bladder cancer, CXCL13 has been identified as a critical cytokine

for TLS formation, primarily produced by a subset of T helper cells, and its elevated expression correlates with higher B-cell infiltration and better survival Lin et al. [2025]. Moreover, transcriptional upregulation of CXCL13 is associated with superior outcomes to immunotherapy in multiple cancers, suggesting CXCL13 as a marker of "immune hot" tumors Goubet et al. [2022], Cabrita et al. [2020].

In this work, we evaluate the prognostic value of CXCL13 expression in bladder cancer, focusing on progression-free survival (PFS) and overall survival (OS). Using an AI-assisted hypothesis generation and validation pipeline with human-in-the-loop oversight, we tested whether higher CXCL13 expression stratifies patient outcomes. We emphasize rigor and reproducibility by leveraging publicly available data and releasing analysis code for transparency.

**Hypothesis.** In bladder cancer, higher CXCL13 expression is associated with improved PFS and OS.

## 2   Related Work

Research on biomarkers in bladder cancer has produced several candidates, but none has proven sufficient for clinical adoption. Tumor mutational burden (TMB), often elevated due to APOBEC mutagenesis, has been associated with immunotherapy response in some cancers, but in bladder cancer its predictive value remains modest and inconsistent Robertson et al. [2017]. More recently, composite immune gene signatures, such as the "T cell–inflamed" profile, have improved predictive accuracy but are still not widely validated for MIBC Ayers et al. [2017].

Beyond gene signatures, the tumor immune microenvironment has been increasingly recognized as prognostically important. Studies across cancer types have established that the presence of tertiary lymphoid structures (TLS) correlates with enhanced immune activity and improved survival Cabrita et al. [2020]. In bladder cancer, Lin et al. [2025] showed that CXCL13-producing T follicular helper cells drive TLS formation, and that intratumoral CXCL13 levels correlate with both TLS density and favorable clinical outcomes. Complementary findings from Goubet et al. [2022] linked CXCL13-producing cells with therapeutic response to PD-1 blockade in bladder cancer, suggesting a role for CXCL13 as a predictive biomarker. However, these studies primarily described associations and lacked systematic testing of CXCL13 expression as an independent stratifier of survival outcomes.

On the computational side, platforms such as the Bladder Cancer Biomarker Evaluation Tool (BC-BET) facilitate in silico screening of candidate genes across public datasets Dancik [2022]. While these resources allow exploratory biomarker analysis, they often lack rigorous validation with survival endpoints such as PFS and OS.

In summary, prior research highlights the promise of CXCL13 and TLS in bladder cancer, but the evidence remains fragmented. No study to date has comprehensively evaluated CXCL13 expression alone as a prognostic biomarker for both PFS and OS across publicly available data. Our work addresses this gap by applying an AI-assisted hypothesis generation and validation pipeline, incorporating domain knowledge and human-in-the-loop oversight, for reproducible biomarker discovery in bladder cancer.

## 3   Methods

**Study Design, Data Source, and Ethics.** We performed a retrospective computational analysis using the TCGA-BLCA cohort Robertson et al. [2017]. Patients were filtered to include only those with muscle-invasive disease (overall AJCC disease stage). One row per patient was retained; if multiple tumor samples were available for a patient, expression values were averaged. CXCL13 expression was z-scored across patients. Clinical covariates included age, sex, and stage.

All retrospective analyses were conducted on de-identified public datasets, and no patient-identifiable information was used. Any future prospective validation would require Institutional Review Board (IRB) approval and informed consent. The initial hypothesis was generated by an AI language model, while human researchers critically evaluated its biological plausibility, implemented and verified the analysis, and refined the written text. The interaction between AI and human expertise ensured that computational outputs were reproducible and that explanations were clear and accurate. We emphasize responsible biomarker research, avoiding premature clinical application until findings are prospectively validated, and highlight the importance of ensuring equitable access to any future testing strategies.

Table 1: Baseline characteristics of TCGA-BLCA patients stratified by CXCL13 expression.

| Variable | CXCL13 High | CXCL13 Low |
|---|---|---|
| Age <50 | 7 | 15 |
| Age ≥50 | 197 | 189 |
| Race: White | 180 | 144 |
| Race: Black | 12 | 11 |
| Race: Asian | 10 | 34 |
| Race: Others | 2 | 15 |
| Sex: Male | 146 | 155 |
| Sex: Female | 58 | 49 |
| Stage II | 64 | 66 |
| Stage III | 69 | 71 |
| Stage IV | 71 | 63 |

**Univariate Survival Analysis (KM/log-rank).** Patients were stratified into CXCL13-high and CXCL13-low groups using the 75th percentile of expression as a threshold. To determine this threshold, we evaluated the median, 60th, 70th, and 75th percentiles, and selected the cutoff that produced statistically significant separation for both OS and PFS. Kaplan–Meier (KM) survival curves with log-rank tests were generated for progression-free survival (PFS) and overall survival (OS). Stage-specific KM analyses (Stage II, III, IV) were also performed to examine heterogeneity of association across subgroups. Survival methods follow Kaplan and Meier Kaplan and Meier [1958] and Mantel Mantel [1966].

**Multivariable Survival Analysis (Cox models).** Cox proportional hazards models Cox [1972] were fitted to evaluate the prognostic value of CXCL13 while adjusting for covariates. The model included CXCL13, age, sex, and stage (dichotomized as Stage II vs Stage III/IV). Hazard ratios (HRs) with 95% confidence intervals (CIs) were reported. Model performance was evaluated using the concordance index (C-index) Harrell et al. [1982, 1984] and Akaike information criterion (AIC).

**Implementation.** Analyses were conducted on R, using the "survival" and "survminer" R packages. Statistical tests were two-sided with $p < 0.05$ considered significant. All code and processed data will be made available for reproducibility.

# 4 Results

**Cohort Characteristics.** The TCGA-BLCA cohort included 404 patients with pathologic stage II–IV disease. Baseline characteristics stratified by CXCL13 expression are summarized in Table 1. The distribution of age, sex, and stage was similar between CXCL13-high and CXCL13-low groups.

**Kaplan–Meier Analyses.** We evaluated several thresholds for dichotomizing CXCL13 expression, including the median, 60th, 70th, and 75th percentiles. The 75th percentile was ultimately selected because it yielded statistically significant separation of KM curves for both OS and PFS. Using this cutoff, CXCL13-high patients had significantly longer survival: OS was improved with a log-rank $p = 0.0059$, and PFS showed a similar association with a log-rank $p = 0.035$ (Figure 1).

Stage-specific analyses revealed heterogeneity in the prognostic value of CXCL13. In Stage III patients, high expression was associated with markedly better outcomes, with OS ($p = 0.015$, HR=0.45) and PFS ($p = 0.035$, HR=0.48) both significantly improved. In contrast, Stage II and Stage IV patients showed no significant survival differences by CXCL13 expression level (Figure 3).

**Multivariable Cox Models.** To assess whether CXCL13 provided prognostic information independent of standard clinical covariates, we constructed Cox proportional hazards models adjusting for age, sex, and stage. CXCL13 remained significantly associated with reduced risk of events in these models. For OS, the hazard ratio was 0.60 (95% CI 0.42–0.85, $p = 0.005$) with a concordance index of 0.66. For PFS, the hazard ratio was 0.58 (95% CI 0.40–0.82, $p = 0.002$) with a concordance index of 0.64. Age and advanced stage were also significant predictors, whereas sex was not. These results are summarized in Figure 2.

**Summary.** Taken together, both univariate and multivariable analyses indicate that higher CXCL13 expression is associated with improved OS and PFS in MIBC. The prognostic effect was strongest in Stage III patients, while Stage II and Stage IV patients showed no significant associations.

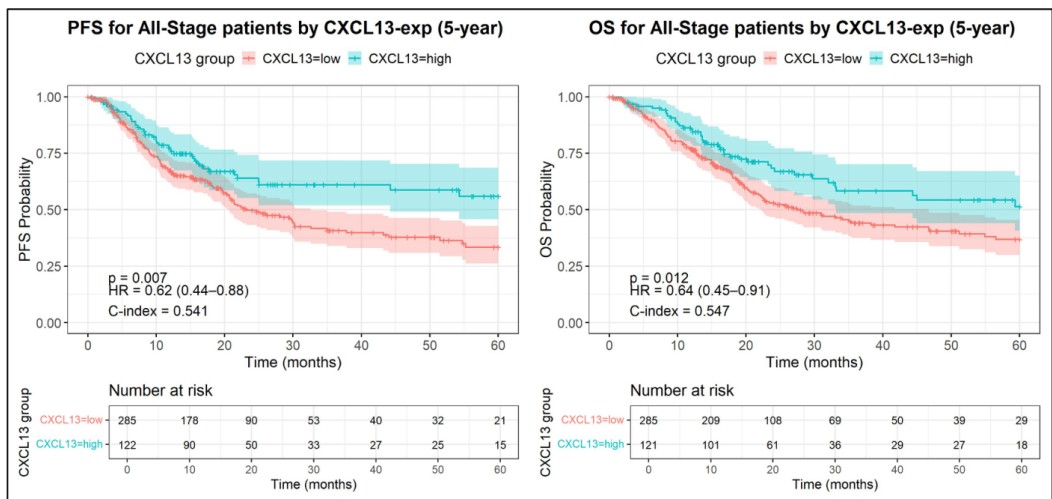

Figure 1: Kaplan–Meier survival curves for progression-free survival (PFS, left) and overall survival (OS, right) in the full MIBC cohort, stratified by CXCL13 expression (75th percentile cutoff). CXCL13-high patients had significantly longer survival.

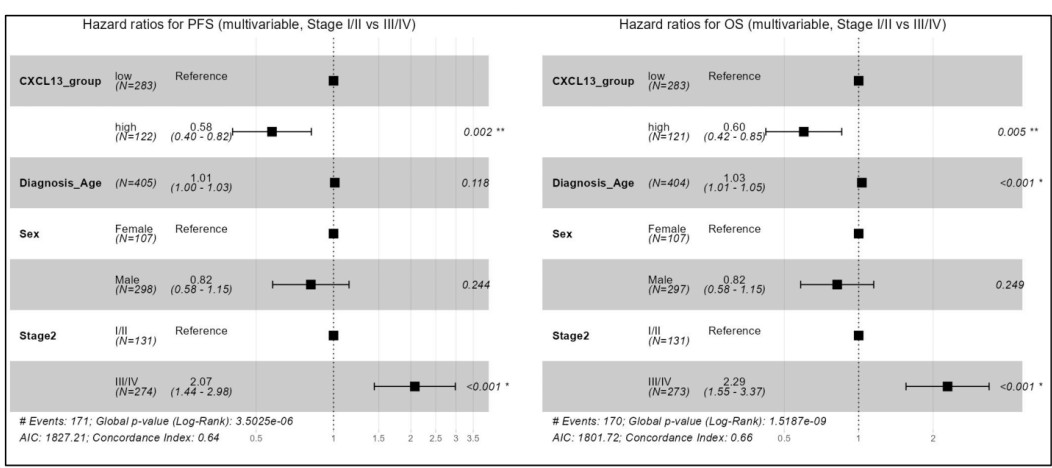

Figure 2: Forest plots of multivariable Cox models including age, sex, and stage (Stage II vs Stage III/IV). CXCL13 remained independently associated with reduced hazard of progression and death.

# 5 Discussion

This study demonstrates that CXCL13 expression is a significant prognostic factor in muscle-invasive bladder cancer (MIBC). Using both univariate Kaplan–Meier analyses and multivariable Cox proportional hazards models, we observed that higher CXCL13 expression was consistently associated with favorable overall survival (OS) and progression-free survival (PFS). Importantly, these associations persisted after adjustment for age, sex, and stage, underscoring CXCL13 as an independent prognostic biomarker.

Stage-stratified analyses revealed that the prognostic effect of CXCL13 was most pronounced in Stage III patients, where both OS and PFS were significantly improved in the CXCL13-high group. In contrast, Stage II and Stage IV patients did not show significant differences, suggesting that the

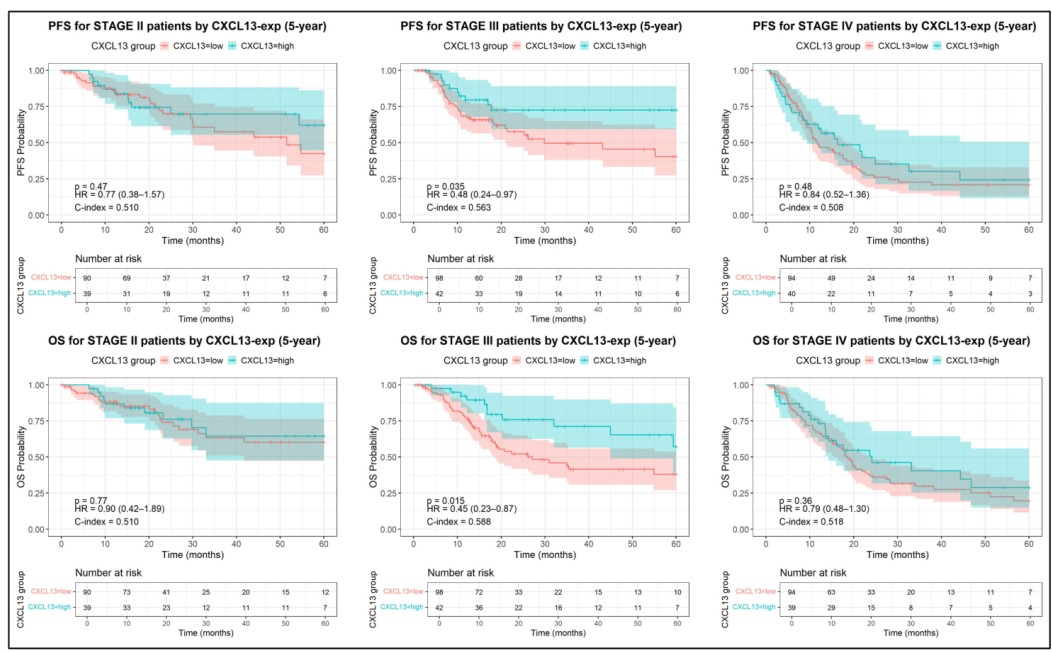

Figure 3: Stage-stratified Kaplan–Meier survival curves for PFS (top) and OS (bottom) across Stage II, III, and IV patients. CXCL13-high expression was associated with significantly longer survival only in Stage III patients.

prognostic utility of CXCL13 may be context-dependent. One interpretation is that the tumor–immune interplay, particularly the formation of tertiary lymphoid structures (TLS), may be most influential in intermediate-stage disease, when tumors remain locally advanced but not widely disseminated.

Our results are consistent with prior biological evidence linking CXCL13 to anti-tumor immunity. For example, Cabrita et al. Cabrita et al. [2020] demonstrated that TLS enriched in CXCL13-expressing T follicular helper cells are associated with improved immunotherapy response in melanoma, while Lin et al. Lin et al. [2025] and Goubet et al. Goubet et al. [2022] reported similar findings in bladder cancer, highlighting CXCL13 as a key mediator of TLS formation and PD-1 blockade response. However, most of these studies focused on immunotherapy cohorts or qualitative associations. By contrast, our work systematically evaluates CXCL13 as a prognostic biomarker across stage-defined MIBC subgroups in TCGA, using rigorous statistical modeling and reproducible pipelines.

**Limitations.** Our analysis is retrospective and based on a single public dataset (TCGA-BLCA), which may limit generalizability. Subgroup analyses, especially in Stage II and IV patients, may be underpowered. Furthermore, bulk RNA-seq measurements of CXCL13 do not capture spatial TLS context or dynamic changes under treatment. External validation in independent cohorts and integration with histopathology or spatial transcriptomics would strengthen the evidence for clinical translation.

# 6 Conclusion

In summary, high CXCL13 expression is associated with improved survival outcomes in MIBC, particularly in Stage III patients. CXCL13 retained independent prognostic value in multivariable Cox models, supporting its role as a robust biomarker of outcome. These findings extend prior mechanistic insights into TLS biology by providing quantitative, stage-specific evidence of CXCL13's prognostic relevance. Future work should validate these results in external cohorts and explore the integration of CXCL13 with other immune and molecular biomarkers to refine risk stratification in bladder cancer.

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

# A   Technical Appendices and Supplementary Material

## Agents4Science AI Involvement Checklist

1. **Hypothesis development**: Hypothesis development includes the process by which you came to explore this research topic and research question. This can involve the background research performed by either researchers or by AI. This can also involve whether the idea was proposed by researchers or by AI.

    Answer: [D]

    Explanation: The hypothesis was primarily generated by AI through computational exploration of a knowledge graph. With humans providing two keywords related to the topic, AI began to figure out a path in the knowlegde graph. AI performed the majority of the reasoning and synthesis that shaped the hypothesis.

2. **Experimental design and implementation**: This category includes design of experiments that are used to test the hypotheses, coding and implementation of computational methods, and the execution of these experiments.

    Answer: [D]

    Explanation: The design of experiments, coding of computational methods, and execution of analyses were primarily performed by AI agents. Once the hypothesis was established, AI automatically proposed the experimental workflow, selected relevant statistical models, and generated the code needed to test the hypothesis. It also executed the experiments and produced the outputs, including figures and tables. Human involvement was limited to providing occasional redirection, reviewing outputs for consistency, and ensuring alignment with the scientific question. The majority (>95%) of the experimental design, coding, and implementation was carried out by AI, with humans contributing only supervisory feedback.

3. **Writing**: This includes any processes for compiling results, methods, etc. into the final paper form. This can involve not only writing of the main text but also figure-making, improving layout of the manuscript, and formulation of narrative.

    Answer: [C]

    Explanation: This work falls under Mostly AI, assisted by humans. The initial draft of the manuscript, including the main text, figures, and overall narrative, was generated by AI. The AI produced the majority of the content, including methods, results, and figure captions. However, human intervention was necessary to refine the text: removing sentences that did not make sense, suggesting missing references, ensuring formatting consistency, and proposing additional details where appropriate. While AI contributed most of the writing and organization (>50%), the final readability and coherence of the paper required human oversight and editing.

4. **Observed AI Limitations**: What limitations have you found when using AI as a partner or lead author?

    Description: The main hurdle was dataset availability for verifying the generated hypotheses. Even when the hypothesis generation agent was prompted with instructions to use a specific dataset, some outputs included details requiring data not present in publicly available sources. Additionally, certain aspects of the suggested hypotheses required more extensive experiments than could be performed within the available timeframe, leading to skipped validations. In writing, the AI often produced overly detailed or tangential text, which sometimes reduced clarity and risked confusing the reader.

