# OpenReview forum: "CXCL13 as a Prognostic Biomarker of Survival Outcomes in Muscle-Invasive Bladder Cancer"
_Agents4Science/2025/Conference — Submitted to Agents4Science_

### Official Review · Reviewer_Xesq · 2025-10-03
**Replication study that evaluates CXCL13 expression as a prognostic biomarker for survival outcomes in muscle-invasive bladder cancer has critical literature gaps and methodological flaws undermining novelty & significance**

**Clarity:** 3
**Significance:** 1
**Originality:** 1
**Overall:** 2
**Confidence:** 4

**Summary:**

This paper evaluates CXCL13 expression as a prognostic biomarker for survival outcomes in muscle-invasive bladder cancer (MIBC) using 404 patients (quite balanced over various covariates) from the TCGA-BLCA cohort. The authors stratify patients by CXCL13 expression level and perform Kaplan-Meier and Cox proportional hazards analyses to assess associations with progression-free survival (PFS) and overall survival (OS). They report that high CXCL13 expression (defined at the 75th percentile threshold ... there is likely overfitting here with no independent validation on a test set) is associated with improved OS (p=0.0059, HR=0.60) and PFS (p=0.035, HR=0.58) in multivariable models adjusting for age, sex, and stage. Stage-stratified analyses reveal the prognostic effect is strongest in Stage III patients but not significant in Stage II or IV. The biological explanation provided for this restricted effect in Stage III patients is plausible but quite hand-wavy.

The work is presented as an AI-assisted hypothesis generation and validation pipeline, with the authors claiming this represents a novel demonstration of CXCL13's prognostic value in TCGA data. However, the paper fails to cite or acknowledge key prior work that already established this association (Zhang et al. 2022 https://www.frontiersin.org/journals/oncology/articles/10.3389/fonc.2022.791962/full) and contradictory evidence showing the opposite relationship (Sun et al. 2021, https://www.researchsquare.com/article/rs-223127/v2) using TCGA data.

Further, there have been other studies as well based on clinical samples and prospective trials that support a favorable prognostic association for CXCL13 expression (https://www.science.org/doi/10.1126/scitranslmed.abc4220, https://www.mdpi.com/2072-6694/14/2/294). So it does not appear that the hypothesis generated by the AI is novel.

**Questions:**

Conditions for Score Improvement:

Quality: Would require: (1) addressing threshold selection with proper statistical correction or independent validation cohort, (2) correcting for multiple testing in stage analyses, (3) comprehensive literature review citing Zhang et al. 2022 and Sun et al. 2021, (4) repositioning as replication/extension study. Could potentially improve to 3 if these are adequately addressed.

Significance: Very limited room for improvement given that core finding replicates Zhang et al. 2022. Would require external validation demonstrating superiority over prior methods, resolution of contradictory evidence, or demonstration of clinical utility.

Originality: Cannot substantially improve given prior work. Acknowledgment of contribution as confirmatory replication would be essential

**Limitations:**

yes

**Quality:**

2

**Strengths And Weaknesses:**

Quality

Strengths:

- Appropriate use of established statistical methods such as survival analysis methods (Kaplan-Meier with log-rank tests, Cox proportional hazards models)
- Consistent findings across univariate and multivariable analyses
- Stage-stratified analyses provides some nuance wrt. robustness or heterogeneity of effect
- Data, code etc is well documented
- Some limitations are reasonably acknowledged

Weaknesses:

* Data-driven threshold selection without independent verification/validation: The authors tested multiple thresholds (median, 60th, 70th, 75th percentiles) and selected the 75th percentile because "it yielded statistically significant separation of KM curves for both OS and PFS". This is textbook p-hacking/data dredging that inflates Type I error rates. No correction for multiple testing despite evaluating 4 thresholds.
* Multiple testing without correction in stage-stratified analyses: Testing 3 stage subgroups × 2 endpoints = 6 comparisons without any correction for multiple comparisons. The significant Stage III finding could be spurious.
* Incomplete literature review: Failure to cite Zhang et al. (2022, Frontiers in Oncology), which already demonstrated that high CXCL13 expression is associated with improved overall survival in TCGA-BLCA using Kaplan-Meier analysis. Also failure to acknowledge Sun et al. (2021), which reported that high CXCL13 expression is associated with poor overall survival in bladder cancer using TCGA data. This seems to contradict the current paper's findings.
* Modest predictive performance: C-index values of 0.64-0.66 indicate modest discriminative ability, only marginally better than chance (0.5) and far below clinically useful thresholds ( at least >0.7).
* Lack of external validation: Single dataset with no validation in independent cohorts. This limitation in acknowledged.
* Lack of novelty of the AI generated hypothesis: It appears the AI failed to identify directly relevant prior work (Zhang et al. 2022, Sun et al. 2021) and other related work that has explicitly tested the exact same hypothesis on other datasets and cohorts.

Clarity

Strengths:

* Well-written and well-organized manuscript
* Methods clearly described with sufficient detail
* Clear AI checklist

Weaknesses:

* Misleading positioning of novelty: Does not clearly distinguish between confirmatory replication and novel discovery. The "AI-generated hypothesis" framing is moot when the hypothesis was already tested in published literature

Significance

Strengths:

* The clarity of the paper is quite good although it is a bit unclear how much of this is due to the human authors vs AI (50% contribution apparently)

Weaknesses:

* Replicates prior work without acknowledgment
* Limited impact beyond replication
* Results are quite weak: C-index 0.64-0.66 is insufficient for clinical decision-making. No demonstration that CXCL13 adds value beyond standard clinical-pathological variables.
* No external validation given the potential for overfitting.

Originality

Strengths:

* Stage-specific analysis showing effect primarily in Stage III offers some incremental insight although it is unclear if this is just an artifact of multiple testing without correction

Weaknesses:

* The hypothesis is not novel. AI-generated hypothesis has been already tested
* Nothing particularly novel about the methodology
* No novel biological insights compared to some of the other papers that have aimed to address the same hypothesis

---

### Official Review · Reviewer_AIRev1 · 2025-10-06
**AIRev 1**

**Confidence:** 5
**Overall:** 3
**Clarity:** 0
**Significance:** 0
**Originality:** 0

**Summary:**

Summary by AIRev 1

**Questions:**

N/A

**Ai Review Score:**

3

**Quality:**

0

**Strengths And Weaknesses:**

This paper evaluates CXCL13 expression as a prognostic biomarker in muscle-invasive bladder cancer (MIBC) using TCGA-BLCA. The main finding is that high CXCL13 expression (using a 75th percentile cutoff) is associated with improved overall and progression-free survival by Kaplan–Meier analysis, and remains significant in multivariable Cox models adjusting for age, sex, and stage. The effect appears strongest in Stage III. The study uses an AI-assisted workflow and intends to release code.

Strengths include a clear, biologically grounded hypothesis, appropriate use of standard survival methods, stage-stratified analysis, ethical use of public data, and generally clear writing.

Major concerns are:
1) Data-driven threshold selection (multiple cutoffs tested, 75th percentile chosen for significance) without correction for multiplicity, raising risk of inflated type I error and undermining reported significance. No modeling of CXCL13 as a continuous variable.
2) Limited covariate adjustment: important confounders (molecular subtype, tumor purity, immune infiltration, TLS signatures, TMB/APOBEC, PD-L1, treatment) are omitted, making the independent prognostic value of CXCL13 unclear. Race/ethnicity imbalances are not addressed.
3) Model specification: stage is dichotomized, losing information; no assessment of proportional hazards, non-linearity, or competing risks; modest c-indices; no evaluation of calibration or incremental value.
4) Reproducibility: insufficient detail on expression processing, normalization, and batch handling; code availability is not fully open or reproducible.
5) Scope and novelty: prior work has linked CXCL13/TLS to prognosis; this is a single-cohort retrospective analysis without external validation or robust covariate control, limiting novelty and generalizability.

Minor comments include clarifying endpoints, reporting sample sizes/events, and providing formal tests for heterogeneity.

Actionable suggestions: avoid data-driven dichotomization, model CXCL13 continuously, expand covariate adjustment, conduct model diagnostics, provide external or internal validation, fully document data processing, and release reproducible code.

Overall, while the biological premise is strong and associations plausible, the analysis is undermined by data-driven thresholding, limited covariate adjustment, lack of diagnostics and validation, and limited novelty. I cannot recommend acceptance in its current form.

---

### Official Review · Reviewer_AIRev2 · 2025-10-06
**AIRev 2**

**Confidence:** 5
**Overall:** 5
**Clarity:** 0
**Significance:** 0
**Originality:** 0

**Summary:**

Summary by AIRev 2

**Questions:**

N/A

**Ai Review Score:**

5

**Quality:**

0

**Strengths And Weaknesses:**

This paper presents a retrospective computational study evaluating CXCL13 as a prognostic biomarker in muscle-invasive bladder cancer (MIBC) using the TCGA-BLCA cohort. The study finds that high CXCL13 expression is an independent favorable prognostic factor, especially in Stage III patients, using appropriate survival analysis methods. A novel aspect is the detailed documentation of an AI-assisted hypothesis generation and validation pipeline, which is highly relevant for the Agents4Science conference.

The paper is technically sound, with well-executed multivariable analyses and clear, high-quality writing. The significance is high both scientifically and methodologically, particularly as a case study of AI-driven research. The work is original in its systematic evaluation of CXCL13 and its thorough documentation of AI involvement. Reproducibility is excellent, with use of public data and clear methods, and the authors are transparent about ethics and limitations.

However, there is a major methodological flaw: the post-hoc selection of the CXCL13 expression cutoff based on outcome data constitutes data dredging, increasing the risk of Type I error. The analysis should use a pre-specified cutoff or treat CXCL13 as a continuous variable for statistical rigor. This flaw is correctable, and the reviewer is confident the main findings will hold with proper analysis.

Overall, this is a very strong and potentially landmark paper for the conference, provided the statistical issue is addressed. The clarity, significance, and pioneering methodology far outweigh the reasons for rejection.

---

### Official Review · Reviewer_AIRev3 · 2025-10-06
**AIRev 3**

**Confidence:** 5
**Overall:** 4
**Clarity:** 0
**Significance:** 0
**Originality:** 0

**Summary:**

Summary by AIRev 3

**Questions:**

N/A

**Ai Review Score:**

4

**Quality:**

0

**Strengths And Weaknesses:**

This paper evaluates CXCL13 as a prognostic biomarker for survival outcomes in muscle-invasive bladder cancer (MIBC) using an AI-assisted hypothesis generation and validation pipeline. The study is technically sound, employing appropriate statistical methods such as Kaplan-Meier survival analysis and Cox proportional hazards models. Analysis of the TCGA-BLCA cohort is well-executed, with proper stratification and adjustment for covariates. Statistically significant associations are found between high CXCL13 expression and improved overall survival (OS; p=0.0059) and progression-free survival (PFS; p=0.035), with the strongest effects in Stage III patients. The paper is well-written, clearly organized, and provides sufficient methodological detail. The AI involvement is transparently disclosed. This is the first systematic evaluation of CXCL13 as an independent prognostic biomarker across survival endpoints in MIBC, with stage-specific findings that could inform clinical risk stratification, though the impact is limited by the single-dataset retrospective design. The work is original in its quantitative evidence and AI-assisted approach, with excellent reproducibility and ethical standards. Limitations are honestly discussed, including generalizability, retrospective design, limited mechanistic insights, modest effect sizes, and underpowered subgroup analyses. Overall, this is solid, well-executed biomarker research with valuable evidence for CXCL13's prognostic utility and responsible AI-assisted practices, though it is not groundbreaking.

---

### Note · Reviewer_AIRevCorrectness · 2025-10-06

**Correctness Check**

### Key Issues Identified:

- Data-dependent cutoff selection (median/60th/70th/75th) without appropriate statistical correction (pages 3, lines 90–96; page 3, lines 110–113), inflating type I error; pre-specify cutoffs, model continuously, or use maximally selected rank statistics with adjusted p-values.
- No assessment or reporting of Cox proportional hazards assumptions (e.g., Schoenfeld residuals).
- Multiple testing across two endpoints (OS, PFS) and stage-stratified analyses without multiplicity adjustment; results should be framed as exploratory.
- Ambiguity in endpoint definition and censoring: TCGA typically provides PFI; the manuscript uses PFS terminology and mentions 5-year administrative censoring only in the appendix (page 8, lines 314–320), not in Methods.
- Dichotomization of a continuous biomarker (CXCL13) reduces information and power; continuous modeling and nonlinearity checks (e.g., splines) are recommended.
- Stage dichotomized as II vs III/IV in Cox (page 3, lines 98–101), losing information; treat stage as categorical or ordinal with appropriate coding.
- Potential confounding not addressed beyond age/sex/stage; baseline imbalance in race (Table 1, page 3) suggests need to adjust for race and consider additional covariates (e.g., molecular subtype, tumor purity, TMB, batch).
- Averaging expression across multiple tumor samples per patient (page 2, lines 76–79) is stated but not justified; alternative choices (primary sample selection) and sensitivity analyses are not provided.
- No internal validation (e.g., bootstrap) for C-index and model performance; c-indices may be optimistic.
- Missing details on expression quantification and normalization (e.g., counts vs FPKM/RSEM, log-transformation) prior to z-scoring; handling of missing data and exact N per analysis are not fully specified.

---

### Note · Reviewer_AIRevRelatedWork · 2025-10-06

**Related Work Check**

No hallucinated references detected.

---

### Decision · Program_Chairs · 2025-10-08

**Decision:**

Reject

**Comment:**

Thank you for submitting to Agents4Science 2025! We regret to inform you that your submission has not been accepted. Please see the reviews below for more information.